# Wastewater Microbiome Analysis for Population Alcohol Abuse

**Jiangping Wu [1], Yan Chen [1], Jiawei Zhao [1], Tanjila Alam Prosun [1], Jake William O'Brien [2], Lachlan Coin [3,4], Faisal I. Hai [1], Martina Sanderson-Smith [5] and Guangming Jiang [1,\*]**

1 School of Civil, Mining, Environmental and Architectural Engineering, University of Wollongong, Wollongong, NSW 2522, Australia; jw130@uowmail.edu.au (J.W.); yc742@uowmail.edu.au (Y.C.); jz943@uowmail.edu.au (J.Z.); tap067@uowmail.edu.au (T.A.P.); faisal@uow.edu.au (F.I.H.)
2 Queensland Alliance for Environmental Health Sciences (QAEHS), The University of Queensland, Brisbane, QLD 4102, Australia; j.obrien2@uq.edu.au
3 Department of Clinical Pathology, The University of Melbourne, Parkville, MEL 3010, Australia; lachlan.coin@unimelb.edu.au
4 Department of Microbiology and Immunology, The University of Melbourne, Parkville, MEL 3010, Australia
5 School of Chemistry and Molecular Bioscience and Molecular Horizons, University of Wollongong, Wollongong, NSW 2522, Australia; martina@uow.edu.au
* Correspondence: gjiang@uow.edu.au; Tel.: +61-02-4221-3792

**Abstract:** This study aims to unveil correlations between wastewater microbiota and the catchment-specific population health risk, specifically alcohol abuse, with smoking and obesity as confounding factors. Our study highlights the importance of extracting human-associated microbial communities from wastewater metagenomes by excluding environmental microorganisms, due to their irrelevance to human health. After excluding environmental microbes, we observed strong associations of all three health risk factors, including alcohol abuse, smoking and obesity, with the human gut microbiome in wastewater. The linear discriminant analysis effect size (LEfSe) analysis showed *Lactococcus_A*, *Leuconostoc*, *Aeromicrobium*, *Akkermansia*, *Weissella*, *Limosilactobacillus*, *Klebsiella_A*, *Desulfovibrio* and *Cloacibacillus* as potential microbial biomarkers for alcoholism, after accounting for the confounding effects of smoking and obesity. Functional annotations of microorganisms linked with lower alcoholism rates are primarily related to energy metabolism and intercellular communication. Microorganisms associated with higher alcoholism rates are predominantly involved in immune regulation and cellular DNA architecture. This study highlights the need for a comprehensive exploration of different health risk factors together to identify potential associations between the wastewater microbiome and population lifestyle.

**Keywords:** wastewater; metagenomics; human gut microbiome; alcohol abuse; LEfSe; microbial biomarker

## 1. Introduction

Alcohol abuse is one of the leading causes of death and disability worldwide, contributing to 3 million deaths and 5% of the global disease burden annually [1]. It is an addictive psychoactive substance that can cause a variety of illnesses, mental and behavioral disorders and damage to others [2]. Considering the high morbidity, mortality and huge economic burden caused by alcohol abuse, monitoring alcohol consumption becomes a crucial foundation for shaping health policy decisions.

When ingested, ethanol is mainly (95–98%) absorbed into the blood circulation through the mucous membranes of the mouth, stomach, and intestines [3]. While the gastrointestinal tract serves mainly as the site of alcohol absorption, prolonged alcohol consumption disrupts the intestinal barrier, erodes the mucosal lining of the upper gastrointestinal tract and disrupts the balance of gut microbiota, such as alterations in *Proteobacteria*, *Faecalibacterium*, *Bacteroidetes* and *Enterococcaceae* [4,5]. Researches also observed prolonged adverse effects of alcohol on gut microbes, which persisted even after one month of abstinence [6].

The relationship between human lifestyle and gut microbiota has been extensively researched. Studies have demonstrated that gut microbes undergo changes in response to varying lifestyles and can reflect the host behavior such as alcoholism [7,8]. However, a substantial portion of microorganisms residing within the human body cannot be identified by culturing methods [9]. Current research predominantly relies on 16S rRNA gene sequencing to discern structural disparities in gut microbial composition between diseased and healthy groups [10]. With the development of high-throughput sequencing, metagenomic sequencing has significantly enhanced the capacity for species identification and functional annotation [11]. In particular, Nanopore sequencing offers long sequence fragments, real-time sequencing and no need for amplification [12].

Domestic wastewater has been used as a public health indicator for so-called wastewater-based epidemiology (WBE) or wastewater surveillance [13]. Human feces has been shown as a substantial contributor to the microbiota in wastewater, with approximately 97% of gut bacterial groups being found in wastewater [14,15]. Differences in the structure of gut microbes were observed in the wastewater from populations with distinct dietary, cultural, and geographical backgrounds [16].

Untreated municipal wastewater incorporates wastewater from households, industry, and the environment, etc. Thus, it is a microbial community that reflects the environmental and other sources in the sewer catchments [17,18]. Some dominant microbes in influent wastewater, including *Arcobacter*, *Acinetobacter*, *Aeromonas*, and *Trichococcus*, were not reported members of the human fecal microbiome [17]. This indicates the necessity to separate the total wastewater microbiome into human and environment-related sub-communities.

Previous studies have demonstrated that microbiological signatures can differentiate populations with alcohol-associated liver disease, making microorganisms relevant for diagnosis and prediction [19,20]. However, microbiota composition may be confounded by other factors like race, body mass index, smoking, etc. [21,22]. It is imperative to consider these confounding effects when searching for alcohol consumption associated microbial biomarkers in wastewater microbiomes. This underscores the necessity of separating the total wastewater microbiome into human-related and environment-related sub-communities, which will be investigated in this study.

This study investigated how wastewater microbiome, a composition of both environment and human-related microorganisms, can be utilized to reveal its association with the community-level alcohol consumption behavior. Nanopore metagenomic sequencing was used to obtain comprehensive microbial community and functional annotations from influent wastewater samples collected from 11 selected wastewater treatment plants with different alcoholism rates in the corresponding sewer sheds. When searching for alcohol consumption-associated microbial biomarkers in wastewater microbiomes, it is essential to consider these confounding effects. This study will explore the confounding effects of smoking and obesity on the microbial biomarkers of alcoholism. We delineated the sources of wastewater microbiome as non-fecal and fecal microorganisms in our analysis. Additionally, we conducted a rigorous LEfSe analysis to search for microbial biomarkers for alcoholism while eliminating the potential confounding impact of obesity and smoking.

## 2. Materials and Methods

### 2.1. Wastewater Sampling

All WWTP catchment samples were selected on their varying alcoholism rate on the 2016 Australian census day. Health statistics for these areas, including alcohol consumption (the percentage of the population aged 18 years and over who consumed more than two standard alcoholic drinks per day), obesity (the percentage of the population aged 18 years and over who were obese), and smoking (the percentage of the population aged 18 years and over who are current smokers) were calculated using data from the Australian Bureau of Statistics based on the population of the service area [23] (Table S2). The prevalence of alcoholism ranges from 4% to 33% across diverse regions in Australia at the Statistical Area Level 2, predominantly concentrated between 10% and 20%. Specif-

ically, the study categorized areas into two groups based on their alcohol consumption rates: a low drinking group (A-Low: 12.70 ± 1.09%) and a high drinking group (A-High: 18.45 ± 2.13%), showing a statistically significant difference ($p < 0.05$). Similarly, obesity ranges from 10% to 60%, primarily concentrated between 20% and 40%. Smoking ranges from 4% to 40%, primarily between 10% and 20%. The low obesity group (24.14 ± 2.28%) differed significantly from the high obesity group (32.22 ± 1.98%). Similarly, the low smoking group (11.04 ± 1.57%) was significantly different from the high smoking group (15.95 ± 1.26%) ($p < 0.05$). In addition, two chemical factors related to smoking and alcohol consumption were analyzed: nicotine (average daily loading rate in mg per 1000 people) and alcohol (standard drinks per 1000 people per day), as reported previously [24].

All influent samples were divided and stored in pre-sterilized polyethylene terephthalate bottles, as outlined in the Supplementary Material [25]. This study encompassed both urban and rural regions across Queensland, New South Wales, and South Australia. Wastewater treatment plants served communities with their population ranging from 80,000 to 1.26 million (Table S1).

### 2.2. DNA Extraction and Nanopore Sequencing

The influent wastewater samples (50 mL each) underwent thorough mixing and subsequent filtration through a 0.22 μm mixed cellulose ester filter (Sigma Millipore, New York, NY, USA). DNA extraction was performed using the Soil FastDNA SPIN kit (MP Biomedicals, Santa Ana, CA, USA). DNA concentrations were quantified using the Quant-iT™ PicoGreen dsDNA Assay Kit (Thermo Fisher, Waltham, MA, USA) with the Qubit 4 Fluorometer.

DNA fragment repair was accomplished using the NEBNext End Repair/DA-tailing Module (New England BioLabs, 7546, Sydney, Australia) and the NEBNext FFPE DNA Repair Mix (New England BioLabs, 6630, Sydney, Australia). Subsequently, ligation-based library preparation was carried out using the Ligation Library Preparation Kit (SQK-LSK-109, Oxford Nanopore Technologies, Oxford, UK). Finally, 75 μL of DNA library was added into one Nanopore GridION flow cell (R9.4) (Oxford, UK) and sequenced for 72 h.

### 2.3. Metagenomic Data Analysis

Following Nanopore sequencing, reads with a quality score of 9 or higher were retained through MinKNOW. Genome assembly was executed utilizing Flye (v.2.9.1-b1780) [26]. Gene mapping was accomplished using minimap2 (v.2.24-r1122) [27], and the results underwent three rounds of polishing and correction via Racon [28].

Open reading frame (ORF) prediction was performed using Prodigal (v.2.6.3) [29], and gene expression abundance was normalized to transcripts per million (TPM, a measure to quantify gene abundance, originally used by the transcriptomics field) using Salmon (v.1.8.0) [30]. For sequence comparison, Blastp was employed against the NCBI taxonomy for total sewage microorganisms and Unified Human Gastrointestinal Genome (UHGG) (version 2.0) databases for human-related microorganisms, with a threshold of $1 \times 10^{-10}$ [31]. The UHGG (Unified Human Gastrointestinal Genome) database collects data on intestinal flora from different populations around the world, while the Non-Redundant Protein Database contains more comprehensive species information. For taxonomic annotation, the UHGG subdivides taxonomic units by adding letter suffixes based on average nucleotide identity (ANI) [32]. The origin of microorganisms, i.e., from human fecal or non-fecal sources, was ascertained by isolation source via BLASTN against the NCBI database, using an e-value threshold of $10^{-5}$ [33]. Finally, functional annotations were conducted by Eggnog-mapper (v.2.1.7), including Cluster of Orthologous Groups (COG), Carbohydrate-Active Enzymes (CAZy), and Kyoto Encyclopedia of Genes and Genomes (KEGG) [34].

### 2.4. Microbiome Analysis for Biomarkers of Alcoholism

Alpha diversity (Shannon and Simpson indices) was calculated using the "vegan" package (v.2.6-4) in R [35]. Associations between the metagenomic microbiome and human health risk factors were assessed based on the relative abundance data (TPM) of the human gut microbiome in wastewater. Pearson correlation coefficients were calculated between relative abundance of microorganisms and different factors to investigate their associations and significance (statistically significant if $p < 0.05$) [36]. These calculations and their visualization were conducted using the "psych" (2.2.9) and ggplot2 (3.4.1) packages in R.

Both NCBI isolation source and UHGG approaches were employed in this study to separate the microorganisms in wastewater into human gut microbiomes. Further analysis based on the co-occurrence correlation network was carried out to reveal the symbiotic relationship of microorganisms in the total wastewater microbiomes. Global Pearson correlations were calculated with the Benjamini–Hochberg procedure to correct the *p*-values and plotted in Gephi (Version 0.10.1) [37].

The Canonical Correspondence Analysis (CCA) implemented in R (R Core Team, 2018) was employed to explore the potential impacts of human health risk factors (i.e., alcohol abuse, smoking and obesity) on the community structure of wastewater microbiomes. LEfSe analysis was executed within a Galaxy framework to search for microbial biomarkers of alcoholism [38]. In the search for alcoholism biomarkers, potential confounding factors including obesity and smoking were accounted for by using them in the subgroup classification (Table S2). LEfSe assesses the effect size of each differential feature through non-parametric statistical tests like the Kruskal–Wallis (KW) and Wilcoxon rank sum tests, followed by effect size analysis using linear discriminant analysis [39]. This approach is widely employed for biomarker discovery in high-dimensional datasets [38].

The identified datasets (CAZy, COG and KEGG) were further analyzed through LEfSe analysis to uncover associations of COGs and KOs to alcoholism.

## 3. Results and Discussion

### 3.1. Microbial Communities in Wastewater Samples

The average sequencing data size for wastewater samples was 11.3 GB (range: 6.9–19.0 GB), with an average N50 of 2.7 KB (range: 1.04–3.85 KB). The data obtained from all samples provided sufficient coverage of the entire spectrum of microorganisms present in the samples. On average, microbial source analysis indicated that about 24% of the microorganisms originated from human feces, which is similar to prior studies (12.1–26%) based on the isolation source in the NCBI database [33,40]. In comparison, the human gut microbiome based on UHGG at the phylum level retained seven out of the top ten microorganisms from the whole wastewater microbiome. This discrepancy on microbial sources was due to the different approaches in the two source tracking methods. In comparison, UHGG, largely based on metagenome-assembled genomes, was shown to yield a 15% higher rate of meta-genotyping as it provides reference genomes of nearly all prevalent prokaryotic species human stool samples from North America or Europe [41]. Figure S1 shows the top 10 most abundant phyla of sewage (SEW) (Figure S1A) and human-related (HUM) (Figure S1B) microorganisms. Due to differences in classification annotation between the UHGG and NCBI databases, the UHGG will classify in more detail based on ANI, but when add all *Firmicutes* of UHGG together, the results are similar compared to NCBI. *Firmicutes*, *Proteobacteria*, *Actinobacteria*, *Bacteroidota*, and *Campylobacterota* are the five predominant phyla among both SEW and HUM microorganisms, accounting for approximately 75% of their total abundance. Consistent with another study, *Proteobacteria*, *Firmicutes*, and *Bacteroidetes* showed high abundance levels [42]. There is considerable overlap at the phylum level, with seven of the top ten compositions being shared between SEW and HUM microorganisms.

The cooccurrence network analysis of SEW indicates that the wastewater microbial community was clearly divided into four distinct clusters (Figure 1). Notably, the microorganisms demonstrating the highest Pearson correlations (Figure S4) are predominantly grouped within the blue circle. Also, microorganisms within the blue, purple, and pink

circles are primarily of environmental relevance. In contrast, microorganisms aggregated within the green circle exclusively comprise HUM microorganisms. It is noteworthy that this aggregation of HUM microorganisms is distinctly separated from the clusters of environment-associated microorganisms.

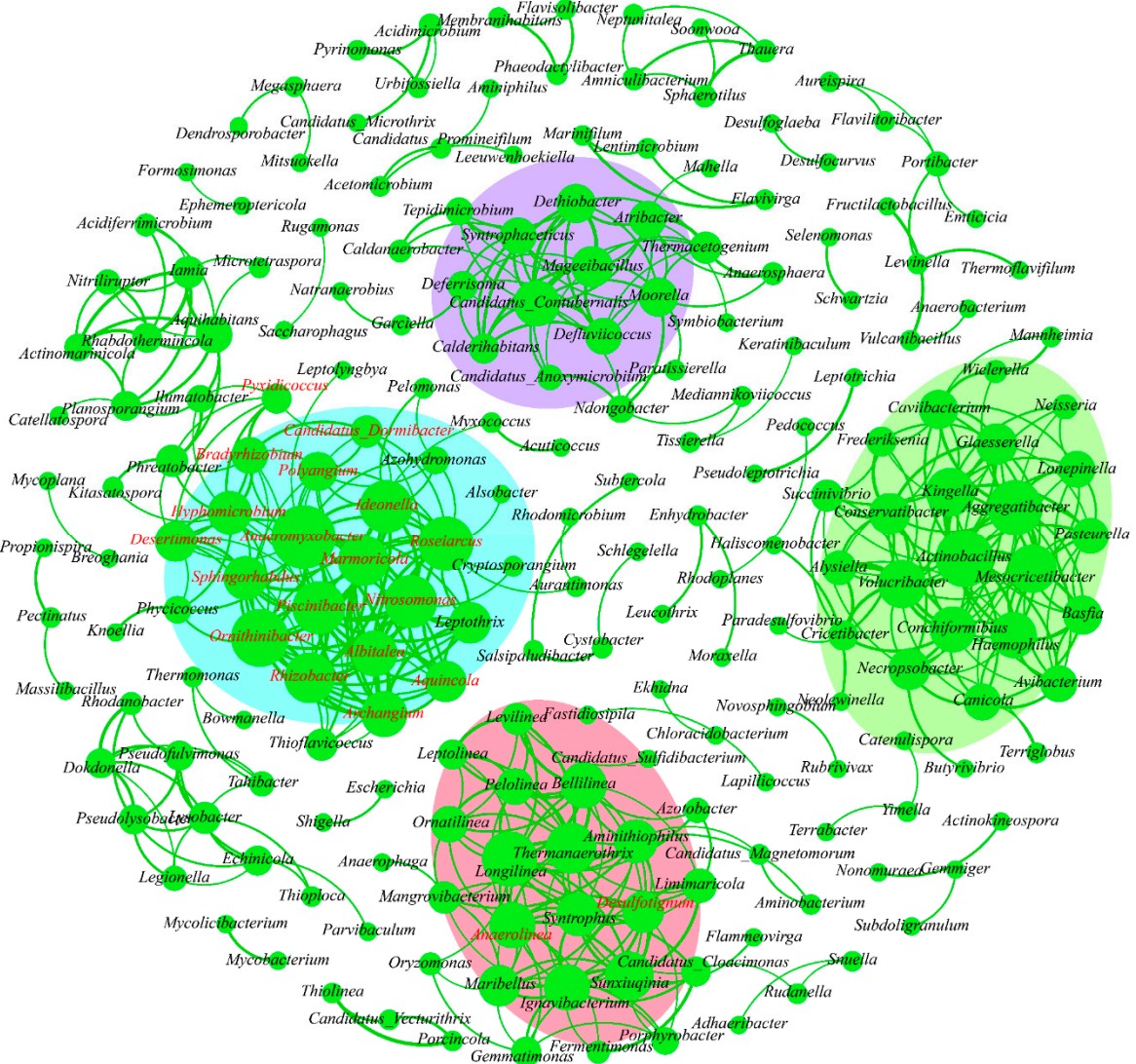

**Figure 1.** The co-occurrence network of genera detected in wastewater ($p < 0.01$, |r| > 0.99). The thickness of connection lines indicates the correlation value, while the node size indicates the importance level. The top 20 most correlated microorganisms are marked in red.

At the genus level, *Streptococcus* and *Blautia* are the two most abundant genera in both SEW (Figure 2A) and HUM (Figure 2B) microorganisms. *Acidovorax* and *Acinetobacter*, as human pathogen were also found to be rich in both categories [43,44]. Previous studies have shown that most fecal microorganisms are detectable in wastewater and play a pivotal role in shaping the wastewater microbial community [33,45]. In addition to human gut bacteria, SEW contains environmental microorganisms such as *Cloacibacterium* and *Aliarcobacter* at 2.17% and 1.89%, respectively [46,47]. Also, microbes originating (*Enterococcus* and *Escherichia coli*) from feces used as the indicator of the contamination were detected in sewage microbiomes [17,33].

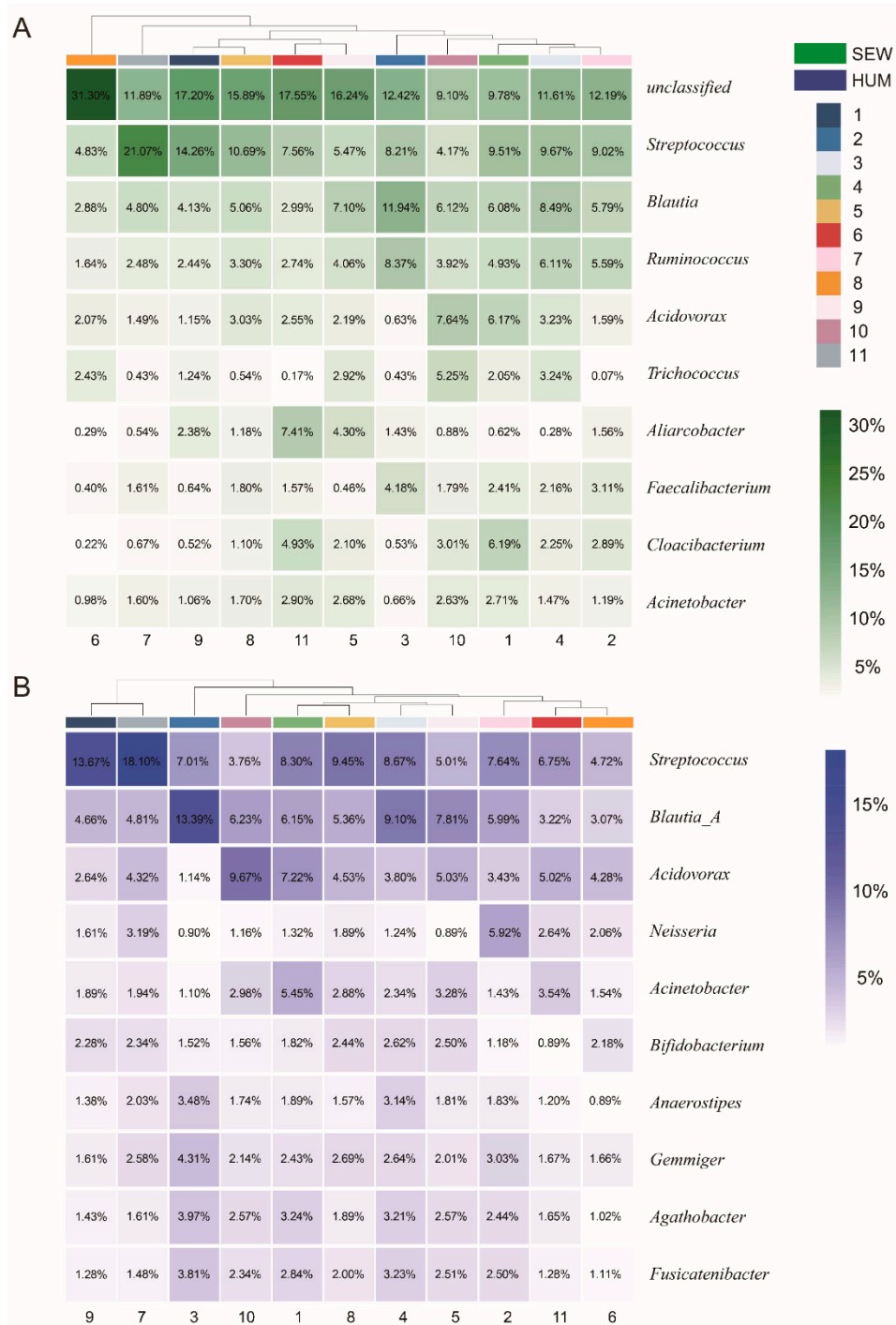

**Figure 2.** Heatmap of sewage (SEW) (**A**) and human-related (HUM) microbial community (**B**) detected in wastewater samples (top 10 most abundant genera).

Figure S2 shows differences in the Simpson (A) and Shannon (B) indexes for SEW and HUM microorganisms at the genus level. Both indexes of HUM are higher than those of SEW. Despite SEW having a higher number of species than HUM, the evenness of species in SEW is reduced when extracting HUM as a separate microbiome, resulting in a decrease in alpha diversity [48].

The $\beta$ diversity analysis of microorganisms was performed using Principal Component Analysis (PCA) (Figure S3), which illustrates the distribution of SEW and HUM in the influent samples. The total wastewater microbiomes (SEW) seem to be clustered based on

sampling geographical locations. Four samples (1, 2, 3, and 10) from NSW are predominantly clustered towards the upper side, while the remaining six samples from QLD are mainly distributed towards the lower side (Figure S3A). The distribution of SEW from NSW and QLD showed substantial dispersion (*p* = 0.001). Also, sample 5 from SA appears isolated on the left side of both QLD and NSW clusters. For HUM microorganisms, the distribution appears also clustered between sampling regions (*p* = 0.025). These findings suggest that the geographic location of the samples may exert a significant influence on the microbial composition.

Previous reports have documented geographical variations in bacterial populations and antimicrobial resistance genes within municipal wastewater [49]. These variations may arise from different temperatures, 48 h precipitation or the influence of wastewater physicochemical parameters, such as flow rates, ammonia levels, biological oxygen demand, total phosphorus, and total suspended solids [50]. In addition, it has been observed that environment-relevant microorganisms can mask the distribution of fecal microorganisms in wastewater [33].

The correlation analysis indicates that the WWTP catchment population and the influent flow rate showed a minimal association with wastewater microbiome (SEW) (*r* = 0.1). Notably, the correlation of the WWTP locations with SEW microbiomes yielded a high coefficient (*r* = 0.43), conforming to the distinct clustering according to geographical regions (Figure S3A).

Interestingly, PCA analysis did not reveal a clear clustering between the low and high alcoholism groups (Figure S3B). This is likely due to the PCA method that primarily captures the largest variance in the dataset, which may not necessarily be related to the differences between the low and high alcoholism groups. These observations also imply that smoking and obesity may cause confounding effects on the microbial community, in addition to alcoholism.

### 3.2. Human Gut Microbiomes Detected in Wastewater

In the HUM microbial community (Figure 2), *Streptococcus* includes pathogenic species commonly found in the nasopharyngeal region, causing respiratory diseases such as pneumonia and septicemia [51]. *Blautia_A* is a microbe that reduces inflammation and maintains intestinal microecology and has been selected as the biomarker of obesity [52,53]. The predominant genera from human gut also include *Fusicatenibacter*, *Bifidobacterium*, *Agathobacter*, and *Anaerostipes*, which are common intestinal microorganisms [54,55]. *Neisseria* is a genus associated with human health, with two pathogenic organisms species [56,57]. *Acidovorax* and *Acinetobacter* are two bacteria prevalent in the environment, but are also human pathogens associated with infections in the blood, urinary tract, and lungs [43,44].

Figure S5 depicts a distinct correlation network analysis for the human gut microbial community. The top 20 human-related genera with the highest number of correlations with each other are visualized in Figure S6, all enclosed within the green circle in Figure 1. The majority of these genera belong to the order *Bacteroidales* (*HGM05376*, *UBA11471*, *W3P20-009*, *OM05-12* and *UBA4372*), a major member of the human gut microbiota [58]. *Bacteroidales* is often employed in fecal source tracking and has been linked to alcoholic liver disease in humans [59]. Prior research has also indicated that *Bacteroidales* can impact intestinal short-chain fatty acid and lipopolysaccharide metabolism, potentially causing obesity in the population [60,61]. *Alistipes*, a recently discovered anaerobic microorganism inhabiting various human diseases, is mostly found in protein-rich dietary groups [62,63]. Additionally, both *Butyricimonas* and *Odoribacter*, SCFA producers crucial for host intestinal health, are from *Bacteroidales* order [64,65]. *Butyricimonas*, an anaerobic microorganism isolated from the intestines of obese patients, exhibits significantly higher levels in the post-drinking population [66,67]. *Brachyspira* is a common enteric pathogen that causes diarrhea [68]. *Parabacteroides* is a major microorganism in the human gut, and was found to be associated with obesity and metabolic syndrome [69]. *Odoribacter* helps reduce obesity-related inflammation [70]. Studies have found that the *Rikenellaceae* family is linked to a

negative association with obesity, a positive association with smoking, and higher levels in abstaining populations which is related to multiple lifestyle habits [71–73]. *Flavobacteriales* (*UBA1820*) and *Haloferacaceae* (*Haloferax*, *Halorubrum*) are prevalent environmental microorganisms. *Flavobacteriales* are known as foodborne pathogens, while *Haloferax* is the first halophilic archaea isolated from humans [74,75].

Among the correlated genera (Table S3), *Jonquetella* and *Acetomicrobium* have the highest correlation (r = 0.998). Both are affiliated with the Synergia family, known for their capacity to produce SCFAs [76]. Our analysis of the human gut microbiota demonstrated a predominance of core microbes specializing in the production of SCFAs. *Pyramidobacter* and *Jonquetella* also exhibit a strong positive correlation (r = 0.998), and research on *Pyramidobacter* has demonstrated an association with rectal cancer [77]. SCFAs have historically played a pivotal role in preserving human health, intimately intertwined with gut microbiota. The abundance of SCFA-producing microbes may correlate with dietary patterns within populations, particularly those favoring high-fiber foods [78].

### 3.3. Impacts of Human Health Risk Factors on Human Gut Microbiome in Wastewater

Figure 3A shows the population per capita drinking rates, based on ethyl sulfate concentration in wastewater, only have a limited correlation (*r* = 0.51) with the alcohol abuse rates. This discrepancy could potentially arise from the fact that the chemical biomarkers mainly reflected alcohol consumption on the sampling date, whereas the alcoholism rate represents the long-term behavioral habits. In comparison, the correlation between smoking rate and nicotine is higher (*r* = 0.74), probably because smokers tend to show steady tobacco consumption behavior with limited daily variations. Interestingly, there is a high correlation between the prevalence of alcoholism and smoking (*r* = 0.71), and between the prevalence of alcoholism and obesity (*r* = 0.64) (Figure 3A). This further supports that the three lifestyle risk factors have confounding effects to each other when investigating their relationship to the human gut microbiomes in wastewater.

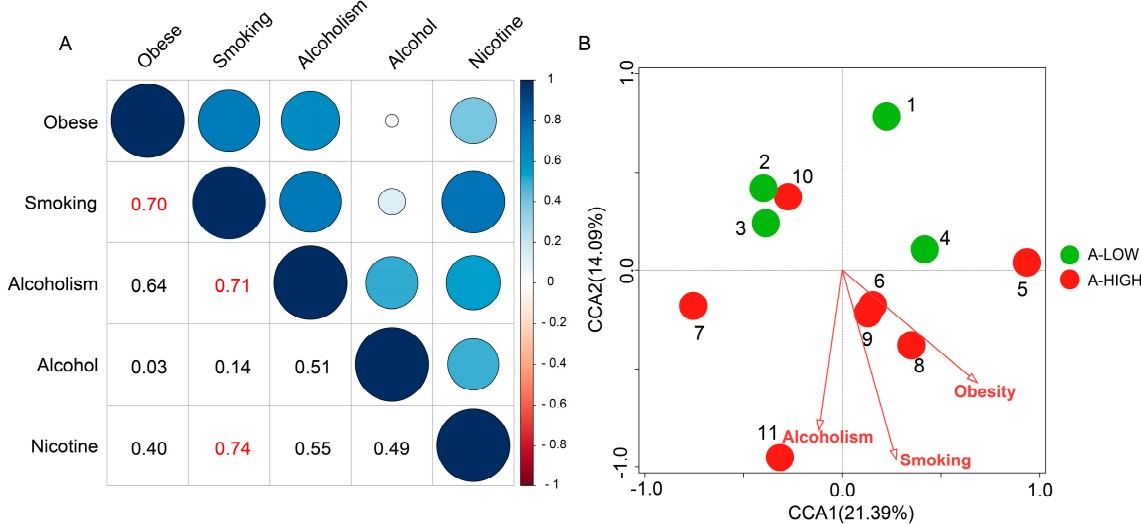

**Figure 3.** (**A**) Correlations between individual human health risk factors (alcoholism, obesity, smoking), and between those and chemical biomarker concentration in wastewater (nicotine for smoking, ethyl-sulfate for alcohol) of 11 selected WWTPs. (**B**) CCA of HUM microorganisms at the genus level. Arrows refer to different health risk factors and the circles represent different WWTPs, in red (high alcoholism rate) or green (low alcoholism rate).

Notably, all three health risk factors, i.e., alcoholism, smoking and obesity, were identified as important impacting factors on the HUM microbiome, accounting for 16.8% (*p* = 0.014), 15.7% and 8.5% of the variance through CCA. Also, the level of obesity and smoking showed similar effects as alcoholism on the HUM microbiomes (CCA2), but

distinct effects on CCA1 (Figure 3B). The arrows for smoking and obesity also point towards similar clusters as alcoholism. It implies that these factors may interact or correlate similarly with microbial communities. However, the different directions projected on CCA1 indicates that the potential confounding effects of obesity and smoking on the wastewater microbiome can be separated from that of alcoholism. It is known that human gut microbial communities are not uniquely shaped by any single health risk factor. Consequently, when assessing the influence of a singular health factor, we took measures to account for these two confounding factors.

*3.4. Potential Microbial Biomarkers of Alcoholism with Smoking and Obesity as Confounding Factors*

We categorized the 11 WWTPs into high and low groups based on regional differences in alcoholism rates ($p = 6 \times 10^{-4}$) (Table S2). Our prior research has shown that population-specific microorganisms in wastewater can effectively differentiate regional populations in relation to obesity and smoking [22,61]. The LEfSe analysis was conducted, with smoking and obesity as important confounding factors, to search for potential microbial biomarkers of alcoholism (Figure 4).

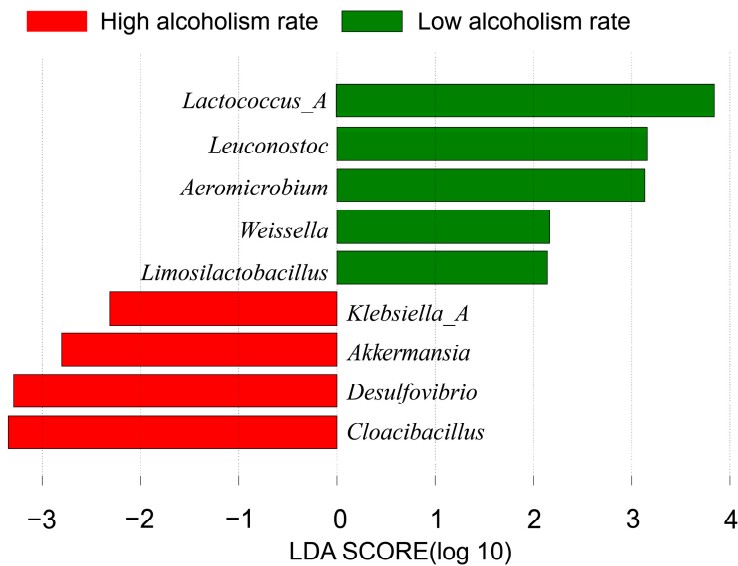

**Figure 4.** Potential microbial biomarkers identified for alcoholism using wastewater microbiomes, via LEfSe analysis ($p < 0.05$, LDA $\geq 2.0$), with obesity and smoking as the confounding factor.

LEfSe analysis identified a total of nine potential biomarkers (Figure 4). Most of them are beneficial bacteria associated with fermentation: *Lactococcus, Lactococcus_A, Leuconostoc, Limosilactobacillus* and *Weissella* [79]. Due to the likely correlation and similar impacts of smoking, obesity and alcoholism on the human gut microbiome, it is imperative to systematically account for and mitigate these confounding effects in the search for alcoholism biomarkers. To identify biomarkers associated with alcoholism, smoking and obesity were employed as secondary classification variables to eliminate their potential confounding effects (Figure S7) [38]. It was observed that mitigating the confounding impact of smoking showed no difference, while removing the confounding influence of obesity resulted in the exclusion of *Lactococcus* (Figure S7B).

After excluding the confounding effects of obesity and smoking on alcoholism, *Lactococcus_A* (*Lactococcus_A raffinolactis, Lactococcus_A piscium_C*), *Leuconostoc, Aeromicrobium, Weissella* and *Limosilactobacillus* were identified as representative of areas with low rates of alcoholism (Figure 4). *Lactococcus*, extensively employed in dairy product fermentation and found within the human gastrointestinal tract, possesses an acetaldehyde dehydrogenase enzyme capable of metabolizing acetaldehyde, thereby contributing positively to alcohol metabolism [80]. Based on the UHGG database, *Lactococcus_A* includes *Lactococcus_A raffinolactis* and *Lactococcus_A piscium_C*, both of which were isolated from human stool samples.

When obesity was used as a secondary classification, *Lactococcus* was no longer considered effective. *Leuconostoc*, a lactic acid bacterium, generates short-chain fatty acids (SCFAs) functioning as host signaling molecules that modulate the host's endocrine system [81]. *Weissella*, once categorized as a lactic acid bacterium suitable for food fermentation [82], has been discovered to possess inhibitory effects against foodborne pathogens [83]. *Limosilactobacillus* is a genus of *Lactobacillus* (LAB) that produces mucus-binding proteins to inhibit the infiltration of pathogenic bacteria [84]. Microorganisms prevalent in regions with high alcohol consumption rates predominantly consist of pathogenic agents associated with diseases. Microorganisms represented in areas with low alcoholism rates primarily consist of lactic acid bacteria.

*Klebsiella_A*, *Akkermansia*, *Desulfovibrio* and *Cloacibacillus* were selected as microbial biomarkers of a high alcoholism rate (Figure 4). Based on the UHGG database, four species belong to *Klebsiella_A*. *Klebsiella*, a prevalent pathogen, is associated with pneumonia and bloodstream infections, particularly noted for its propensity to induce pneumonia in alcohol-abusing patients [85]. *Akkermansia*, an emerging probiotic, has demonstrated the ability to reinforce intestinal barrier integrity and mitigate inflammation [86]. It is associated with alcohol intake, decreased in individuals with alcoholic liver disease, and has potential to mitigate alcohol-induced depression [86,87]. *Desulfovibrio* and *Cloacibacillus* are both pathogens that can cause bacteremia [88,89]. These microorganisms potentially influence alcohol metabolism by generating metabolites like lactic acid and may also mitigate alcohol-induced intestinal inflammation.

The analysis reveals distinct effects when using obesity as a secondary classification. Notably, *Lactococcus* was excluded solely in the case of obesity. This indicates a significant interaction between the degree of obesity and alcoholism. In contrast, smoking appears to have a less pronounced effect. Furthermore, certain biomarkers are similarly affected by both smoking and alcoholism, suggesting they are common markers for both conditions. When smoking was used as the primary classification, 50 differential biomarkers were identified (LDA $\geq$ 2). In contrast, when obesity and alcoholism were applied as secondary categorizations, the numbers of markers identified were 40 and 49, respectively. Furthermore, using obesity as the primary classification yielded 58 differential markers. When smoking and alcoholism were used as secondary classifications under the obesity primary classification, we identified 57 and 58 markers, respectively. This underlines the strong influence of obesity on the classification of smoking and suggests a commonality of markers between smoking and alcoholism. However, this observation might also stem from a lack of sufficient secondary groups within each primary classification (high/low alcoholism) when these secondary classifications are applied.

### 3.5. Microbiological Function and Metabolism in Relation to Alcohol Consumption

Functional analysis demonstrates that alcoholism impacts the metabolic capacity for carbohydrates of population-associated microorganisms in wastewater. Table S4 illustrates the differential functionality of human gut microorganisms in wastewater, identified by LEfSe with obesity and smoking as confounding factors, in regions with varying levels of alcoholism ($p < 0.05$, LDA $\geq$ 2).

Functionally annotated analyses of the CAZy database revealed that GH37 ($\alpha$, $\alpha$-trehalase) and GH38 ($\alpha$-mannosidase) corresponded to low alcoholism rates, whereas GH33 (sialidase) was indicative of regions with a high rate (Table S4). This observation persisted after accounting for confounding factors such as smoking and obesity. Trehalase hydrolyzes the trehalose into glucose, serving as a source of energy for physiological processes within the body [90]. Mannosidase participates in Golgi-associated glycoprotein modification and degradation, playing a pivotal role in intercellular recognition and signaling [91]. Sialidase is localized on the cell surface and participates in diverse intercellular signaling pathways, contributing significantly to immune modulation [92]. High alcoholism may stimulate the metabolic activity of immune-related enzymes within the gut microbiota of the population.

A total of 4441 COGs were identified through COG functional classification. Among these, COG3316 (transposase), COG5527 (protein involved in initiation of plasmid replication), COG2963 (Transposase InsE and inactivated derivatives), COG0636 (type ATP synthase, membrane subunit c/Archaeal/vacuolar-type H+-ATPase, subunit K) and COG0198 (ribosomal protein L26) were associated with a low alcoholism rate, while COG1167 (DNA-binding transcriptional regulator, MocR family, contains an aminotransferase domain) was associated with a high alcoholism rate ($p < 0.05$, LDA $\geq$ 2, Table S4). Among them, COG3316, COG5527 and COG2963 all belong to the classification L (replication, recombination and repair), and COG0636 belongs to energy production and conversion. While COG1167 belongs to transcription.

K07498, K02110 and K02895 were selected to represent areas with low drinking rates through the KEGG annotation ($p < 0.05$, LDA $\geq$ 2). K07498 and K02895 are both categorized in the realm of genetic information processing. K02110 on the other hand, belongs to energy metabolism. The gene K01972, chosen to indicate increased alcohol consumption, was also grouped within the genetic information processing category. Immune disease was screened out by areas with high alcohol prevalence on the KEGG L1.

Functional annotation indicates that microorganisms associated with low alcoholism rates primarily engage in energy metabolism and intercellular information transfer, whereas those in regions with high alcoholism rates are mainly related to immunomodulation and cellular DNA structure. These findings suggest that increasing alcoholism rate directly impacts the human health status by influencing immune system dynamics and genetic integrity [93].

### 3.6. Implications and Limitations

The findings of this study reveal that sewage predominantly harbors non-fecal microorganisms, which exhibit intricate interdependencies among themselves. These complex relationships appear to be influenced by factors like the sampling location and other environmental variables. Notably, the coexistence of non-fecal microorganisms introduces significant noise when attempting to directly glean insights into human health from the total sewage microbiota. This highlights the impact of environmental conditions on non-fecal microorganisms, underscoring the challenge of isolating the human-related microbiota for the study of population health [94].

The composition of gut microbial communities in populations was reported to be shaped by human habits [42]. This shed light on the use of microbial biomarkers for alcoholism. The conventional chemical biomarker like ethyl sulfate can only indicate recent alcohol consumption, and differ from demographic data that represent long-term drinking habits [95]. However, the microbial biomarkers are susceptible to multiple human behaviors or unknown confounding factors. This study explored how major health risk factors, i.e., smoking and obesity, affect the potential biomarkers of alcoholism. Previous studies have explored differences in gut microbes between alcoholic and non-drinking people [20]. Biomarkers vary across different investigations, likely due to the geographical locations or other living habits related to their health. Comprehensive data are necessary to validate biomarker consistency, particularly when targeting populations with diverse habits from different regions of the world.

However, this study has its limitations because of the small sample size and incomplete investigation of all potential confounding factors. It is essential to investigate more confounding factors, such as disease prevalence and comprehensive socioeconomic and demographic variation within the population. Future investigations should incorporate larger sample cohorts from diverse seasons, watersheds, and sewer configurations. Establishing quantitative relationships through identified biomarkers will be imperative to enable back-calculations of WBE.

### 4. Conclusions

This study employed metagenomic sequencing to explore the associations between human gut microbiome in wastewater and human health risk habits across various sewage samples. The main conclusions are as follows:

- Environmental microorganisms constitute a significant portion of the sewage microbiota, with their composition primarily shaped by geographic locations.
- Major human health risk factors including alcoholism, smoking, and obesity collectively shape the human gut microbiota in wastewater and exhibit correlations between obesity, smoking and alcoholism.
- Several potential biomarkers for alcoholism have been identified using LEfSe, when accounting for confounding factors such as obesity and smoking. These biomarkers include *Lactococcus_A*, *Leuconostoc*, *Aeromicrobium*, *Akkermansia*, *Weissella*, *Limosilactobacillus*, *Klebsiella_A*, *Desulfovibrio* and *Cloacibacillus*. Microbial biomarkers for alcoholism primarily comprise lactic acid bacteria, which produce lactic acid and influence alcohol metabolism.
- Function annotations of the wastewater metagenomes indicate that alcoholism has the potential to enhance host energy metabolism and intercellular signaling while simultaneously compromising the overall immune function in the host organism.

**Supplementary Materials:** The following supporting information can be downloaded at: https://www.mdpi.com/article/10.3390/w16152149/s1, Figure S1: Top ten most abundant phyla in microbial community detected in wastewater samples (A) and separated human gut microbiome using UHGG database from the same wastewater samples (B); Figure S2: The alpha diversity of sewage (SEW) and human-related (HUR) microbial community in sewage samples. Figure S3: Results of Principal Component Analysis (PCA) of microbial composition in wastewater samples at genus level. Figure S4: Correlation-heatmap for the top 20 genera in wastewater with the strongest relationships based on global Pearson correlations. Figure S5: The cooccurrence network of human gut genera detected in wastewater ($p < 0.01$, $|r| > 0.96$). Figure S6: Correlation-heatmap for the top 20 human related genera with the strongest relationships based on global Pearson correlations. Figure S7: Protentional HUM biomarkers screened for alcoholism ($p < 0.05$, LDA $\geq$ 2.0). Obesity (A) and smoking (B) were used as subgroups. Table S1: Summary of basic information on sampled WWTPs.; Table S2: Census information on sampled WWTPs. Table S3: The top 20 most inter-correlated (two-by-two) microorganisms related with human in the sewage samples ($p < 0.05$). Table S4. Functional characterization of HUM microbial communities in sewage screened by LEfSe ($p < 0.05$, LDA $\geq$ 2.0). The genes in red are shared. Reference [96] is cited in the Supplementary Materials.

**Author Contributions:** Methodology, M.S.-S.; Validation, T.A.P.; Investigation, J.W., Y.C. and J.Z.; Data curation, J.W.; Writing—original draft, J.W.; Writing—review & editing, J.W.O., L.C. and F.I.H.; Supervision, G.J.; Project administration, G.J. All authors have read and agreed to the published version of the manuscript.

**Funding:** This research was funded by Australian Research Council Discovery Project (DP190100385).

**Data Availability Statement:** The datasets presented in this article are not readily available because the data are part of an ongoing study. Requests to access the datasets should be directed to the author.

**Acknowledgments:** J.W. received support from a PhD scholarship from the University of Wollongong. Y.C. received the financial support from the China Scholarship Council and an International Postgraduate Tuition Award (IPTA) from University of Wollongong. Jake W. O'Brien is the recipient of an NHMRC Emerging Leadership Fellowship (EL1 2009209). The samples were collected as part of an Australian Research Council Linkage Project (LP150100364), and the authors wish to thank the QAEHS staff and students who conducted the sampling campaign and the wastewater treatment utilities who provided the samples and metadata. The Queensland Alliance for Environmental Health Sciences, The University of Queensland, gratefully acknowledges the financial support of the Queensland Department of Health.

**Conflicts of Interest:** The authors declare no conflict of interest.

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
