# Peer review of "Wastewater Microbiome Analysis for Population Alcohol Abuse"

_water, doi:10.3390/w16152149_

Round 1

Reviewer 1 Report

Comments and Suggestions for Authors

The manuscript with the title “Wastewater Microbiome Analysis for Population Alcohol Abuse” screened the wastewater microbiome, and the characterization showed microbiological indicators potentially associated with gut microbial population in alcoholic population. The current interest in gut microbiome is very high, because there is evidence that lifestyle and health conditions are heavily intertwined with gut microbiome.

The innovative aspect of the manuscript is that it provides primary evidence that wastewater microbiota can be monitored to provide information on the lifestyle choices of the population that can negatively impact their health.

Best regards.

Author Response

Comments:

The manuscript with the title “Wastewater Microbiome Analysis for Population Alcohol Abuse” screened the wastewater microbiome, and the characterization showed microbiological indicators potentially associated with gut microbial population in alcoholic population. The current interest in gut microbiome is very high, because there is evidence that lifestyle and health conditions are heavily intertwined with gut microbiome.

The innovative aspect of the manuscript is that it provides primary evidence that wastewater microbiota can be monitored to provide information on the lifestyle choices of the population that can negatively impact their health.

Response: We are grateful to the reviewer about the positive comments made to the manuscript. We appreciate the time and effort you have dedicated.

Reviewer 2 Report

Comments and Suggestions for Authors

Dear authors/editors,

I have reviewed the manuscript entitled “Wastewater Microbiome Analysis for Population Alcohol Abuse” wherein the authors aim to discover relationships between wastewater microbiota and community health risks like alcohol abuse. In my opinion the manuscript is well structured, the investigation makes sense and it is worth of investigation. The indirect community issues indicators in wastewater are very interesting to get ahead of problems and therefore this work is very interesting in my opinion. However, I could find some drawbacks to improve, please find them below.

-There are typos in the manuscript such as the end of the authors reading “and *”. Please revise.

-The acronym HUM is not very suitable for “human gut related” in my opinion. Could it be HGR or something like this? HUM is confusing.

-In figure 3A, you do not have any chemical biomarker for obesity so, is obesity a good factor in that table? Does it have any sense?

-Moreover in figure 3A you should correlate the risks with the biomarkers as you state in the caption. In the figure you correlate the risks among themselves.

-In figure 4 “alcohol-high” and “alcohol-low” are not sufficiently selfexplicative for a leyend. Or change them into something like “high alcoholism areas” or explain both in the caption.

-I would have loved to read section 3.6 at the beginning, so my mind could have considered all that while reading the manuscript instead of changing my mind at the end with some facts. I am not sure where to locale it better, but consider explaining it in an earlier part of the manuscript.

-Could you discuss if this study is valid for other parts of the world with different habits but same human risk factors? In my opinion it is an interesting discussion independently of the answer.

Comments on the Quality of English Language

-There are typos in the manuscript such as the end of the authors reading “and *”. Please revise.

Author Response

Comments:

I have reviewed the manuscript entitled “Wastewater Microbiome Analysis for Population Alcohol Abuse” wherein the authors aim to discover relationships between wastewater microbiota and community health risks like alcohol abuse. In my opinion the manuscript is well structured, the investigation makes sense and it is worth of investigation. The indirect community issues indicators in wastewater are very interesting to get ahead of problems and therefore this work is very interesting in my opinion. However, I could find some drawbacks to improve, please find them below:

We are grateful to the reviewer about the positive comments made to the manuscript

Comments 1: There are typos in the manuscript such as the end of the authors reading “and *”. Please revise.

Response: Accepted and changes have been made.

Comments 2: The acronym HUM is not very suitable for “human gut related” in my opinion. Could it be HGR or something like this? HUM is confusing.

Thank you for your comment. We apologize if our description caused any confusion. Our intention was to refer to microorganisms associated with humans. Microorganisms found in sewage are primarily derived from the intestinal tract (feces), but they also include those from the oral cavity and skin, all of which are human-associated. We have removed the term "human gut-related" to "human related".

Line 189 and 219

Comments 3: In figure 3A, you do not have any chemical biomarker for obesity so, is obesity a good factor in that table? Does it have any sense?

Response: Thank you for your question. Obesity is a complex health issue, and there isn't a single chemical indicator that can fully characterize it. However, numerous studies, including population-based experiments, have demonstrated significant differences in gut microbes between obese and normal populations [1]. Some biomarkers have also been identified in this context. Moreover, obesity is frequently linked with various diseases, underscoring its significance as a public health indicator worthy of attention.

Figure 3A shows the correlations between the three different lifestyle and health risk factors. Interestingly, there is a high correlation between the prevalence of alcoholism and obesity (r = 0.64). This further supports the idea that the three lifestyle risk factors have confounding effects on each other when investigating their relationship to the human gut microbiomes in wastewater. 

Comments 4: Moreover in figure 3A you should correlate the risks with the biomarkers as you state in the caption. In the figure you correlate the risks among themselves.

Response: There is a confusion about the figure and its purpose. This figure indeed shows the correlations between health/lifestyle risk factors and biomarkers. On top of that, we also show the correlations between individual lifestyle factors. As indicated above, the high correlations between alcoholism and obesity supports the further analysis of the confounding effects when identifying microbial biomarkers.

Thank you for your question. Changes have been made to clarify the figure.

Line 310-312:(A) Correlations between individual human health risk factors (alcoholism, obesity, smoking), and between those and chemical biomarker concentration in wastewater (nicotine for smoking, ethyl-sulfate for alcohol) of 11 selected WWTPs.

Comments 5: In figure 4 “alcohol-high” and “alcohol-low” are not sufficiently self-explicative for a legend. Or change them into something like “high alcoholism areas” or explain both in the caption.

Response: Thanks for the suggestion, changes have been made to Figure 4.

Comments 6: I would have loved to read section 3.6 at the beginning, so my mind could have considered all that while reading the manuscript instead of changing my mind at the end with some facts. I am not sure where to locale it better, but consider explaining it in an earlier part of the manuscript.

Response: The introduction of the manuscript has discussions related to the complexity of the microbial composition of the wastewater microbiome, which is the driver we are looking at the difference between environment and human related microbiome in the wastewater. Also, the introduction has discussed the potential confounding effects of different lifestyle and health risk factors. We have made some changes to highlight this information so that can be more obvious to the readers of the manuscript.

Line 77: This underscores the necessity of separating the total wastewater microbiome into human-related and environment-related sub-communities, which will be investigated in this study.

Line 85: When searching for alcohol consumption-associated microbial biomarkers in wastewater microbiomes, it is essential to consider these confounding effects. This study will explore the confounding effects of smoking and obesity on the microbial biomarkers of alcoholism.

Comments 7: Could you discuss if this study is valid for other parts of the world with different habits but same human risk factors? In my opinion it is an interesting discussion independently of the answer.

Response: Thanks for the suggestion, this is an intriguing topic for discussion. Discussions have been added.

Line 452-457: Previous studies have explored differences in gut microbes between alcoholic and non-drinking people [2]. Biomarkers vary across different investigations, likely due to the geographical locations or other living habits related to their health. Comprehensive data are necessary to validate biomarker consistency, particularly when targeting populations with diverse habits from different regions of the world.

References

  1. Newton, R.J.; McLellan, S.L.; Dila, D.K.; Vineis, J.H.; Morrison, H.G.; Eren, A.M.; Sogin, M.L. Sewage reflects the microbiomes of human populations. MBio 2015, 6, 10.1128/mbio. 02574-02514.
  2. Zafari, N.; Velayati, M.; Fahim, M.; Maftouh, M.; Pourali, G.; Khazaei, M.; Nassiri, M.; Hassanian, S.M.; Ghayour-Mobarhan, M.; Ferns, G.A. Role of gut bacterial and non-bacterial microbiota in alcohol-associated liver disease: Molecular mechanisms, biomarkers, and therapeutic prospective. Life Sci. 2022, 305, 120760.